# Prevalence and Genomic Diversity of Feline Leukemia Virus in Privately Owned and Shelter Cats in Aburrá Valley, Colombia

**DOI:** 10.3390/v12040464

**Published:** 2020-04-20

**Authors:** Carolina Ortega, Alida C. Valencia, July Duque-Valencia, Julián Ruiz-Saenz

**Affiliations:** Grupo de Investigación en Ciencias Animales—GRICA, Facultad de Medicina Veterinaria y Zootecnia, Universidad Cooperativa de Colombia, Sede Bucaramanga 680005, Colombia; caroortega17@gmail.com (C.O.); alidaval23@hotmail.com (A.C.V.); julyduva@gmail.com (J.D.-V.)

**Keywords:** subgroup classification, feline viral leukemia, retrovirus

## Abstract

The feline leukemia virus (FeLV) belongs to the family Retroviridae; it is the first feline retrovirus discovered and one of the agents that has a great impact on cats’ health and the ecology of the feline population worldwide. It is associated with the occurrence of several syndromes of fatal diseases, including the development of lymphomas. Studies on FeLV have been reported in Colombia, and most of them have been approached from a clinical point of view. However, only a few studies have focused on the prevalence of the infection, while none have clarified which variant or FeLV viral subgroup is presently circulating in our country. Therefore, the present study investigated the prevalence of the infection associated with the molecular characterization of FeLV present in cats in Aburrá Valley, Colombia. The sampling of privately owned and shelter cats was performed in female (*n* = 54) and male (*n* = 46) felines; most of them were seemingly healthy according to the owner’s report, with nonspecific clinical history. Immunoassay confirmed that 59.44% (95% confidence interval (CI) = 49.81–69.06%) of felines were FeLV seropositive. The molecular testing of felines using reverse transcription–polymerase chain reaction and sequencing showed that 30% (30/100) of felines were positive, and the most prevalent subgroup in the Aburrá Valley was FeLV-A. In conclusion, the frequency of leukemia virus, as revealed by molecular and serological tests, is one of the highest reported frequencies to date, and a high molecular variation is shown in the Colombian population. More studies on the behaviour of the virus in feline populations in Columbia are warranted to determine its prevalence throughout the country.

## 1. Introduction

Feline viral leukemia, which is caused by feline leukemia virus (FeLV), was first reported in 1960 by William Jarrett; it is clinically characterized by the development of multiple severe clinical syndromes and is associated with decreased life expectancy in infected cats [1].

FeLV belongs to the order Ortervirales, family Retroviridae, subfamily Orthoretrovirinae, and genus *Gammaretrovirus*. FeLV comprises a single-stranded RNA genome that becomes associated with the target cell through the fusion of the viral envelope with the cell membrane. It then releases the nucleocapsid with viral RNA into the cytoplasm, which is transcribed by the reverse transcriptase (RT) enzyme to DNA and then transported to the cell nucleus, where it is subsequently integrated into the cell genome to become a “provirus”. During cell mitosis, the daughter cells inherit the provirus, indicating that the cat may remain infected for life in such cases [1,2].

The theory that FeLV originated from a rodent-derived virus that evolved to infect cats as a consequence of the predator–prey relationship has been accepted [3]. In addition, exogenous (horizontally transmissible/infectious) FeLVs have been classified into subgroups based on their functional and genetic relatedness. FeLV-A, FeLV-B and FeLV-C subgroups were characterized using viral interference assays and were associated with specific clinical phenotypes. Presently, six subgroups have been classified as follows: A, B, C, D, a fifth group associated with T lymphocytes (defined as FeLV-T and FeLV-TG35-2), and a sixth group that has been recently described (2018), which is genomically related to FeLV-A [4].

Of all the subgroups mentioned, FeLV-A is the only one that can be naturally contagious from cat-to-cat and has been associated with macrocytic anaemia, immunosuppression, and lymphoma [5]. FeLV-B originates through the recombination of endogenous FeLV endogenous retrovirus long terminal repeat (LTR) with FeLV-A and occurs in approximately 50% of the cats infected with FeLV-A. It is tumorigenic and incapable of horizontal transmission unless it is co-transmitted with FeLV-A [4]. It can be affirmed that this recombination process has been tested both in vivo and in vitro in experimental models [6,7]. FeLV-C, FeLV-T, and FeLV-TG35 have focal insertions, substitutions, and deletions within the main FeLV-A virus in different regions. FeLV-C is a less frequently found subgroup that arises from de novo mutations in the env gene of FeLV-A and has been associated with the development of aplastic anaemia. FeLV-D was concurrently identified with the discovery of a novel domestic cat endogenous retrovirus (ERV-DC), which is divergent from the classic endogenous FeLV [8]. FeLV-T, which receives its name owing to its exclusive T-lymphocyte tropism, causes a strong cytopathic effect (the other subgroups are not cytopathic and exit the host cell by budding) [9]; for its effective replication, FeLV-T requires two host cell proteins in the T lymphocytes [10]. FeLV-TG35-2 targets a different receptor, potentially constituting a new subgroup [4].

The result of infection in positive animals varies considerably and includes fever, lethargy, loss of appetite, and weight loss. Infected cats may present with several clinical manifestations simultaneously [11]. Overall, 25% of infected animals present with anaemia because the virus can infect the red bone marrow, causing a reduction in red blood cell count or the abnormal production of erythrocytes that do not function properly, whereas, in other cases, there is a self-destruction by the cat’s own immune system caused by the virus; 15% of infected cats develop cancer, the most frequently occurring type being lymphosarcoma, which causes solid tumours (observable in various sites such as the intestine, kidneys, eyes, and nasal cavity) or leukemia [12]. Although periodontal disease has been reported in felines with FeLV, studies on cases and controls have failed to identify an association between gingivostomatitis and the presence of FeLV antigens [13,14].

The result of FeLV infection is quite different in every cat. Although it primarily depends on their immunity status and age, the outcome is also affected by the strain’s pathogenicity, infecting virus subtype, and the viral concentration to which the susceptible animal is exposed [11]. In addition, a higher mortality rate has been reported for privately owned infected animals coexisting with multiple cats (mortality rate of 50% in 2 years and 80% in 3 years) compared with cats that are well cared for and strictly kept inside homes where there is only one cat [15]. However, the life expectancy of an infected cat with FeLV is still 2.5 times lower than that of an uninfected cat. Studies on cases and controls performed in more than 1000 cats infected with FeLV in the United States compared with more than 8000 control uninfected cats found that the median survival duration was 2.4 years for FeLV-infected cats compared with 6.0 years for uninfected cats [16].

Although several classifications or stages of FeLV infection in domestic cats have been accepted for years, the use of modern methods for diagnosis and the combination thereof have led us to identify four different outcomes of the infection, whose clinical relevance and epidemiological roles are yet to be clarified. The present classification of infection states by FeLV is as follows: abortive infection (comparable to the former "regressor cats"), regressive infection (comparable with the old “transient viremia” followed by “latent infection”), progressive infection (comparable to the former “persistent viremia”), and focal or atypical infection [7].

Studies conducted some years ago in the city of Bogotá in Colombia initially showed prevalence rates ranging between 4% and 13% in the sampled animals [17,18] using the lateral flow immunoassay test for the detection of the viral capsid p27 antigen. A study conducted by the University of Córdoba using the SNAP Combo FeLV Ag / FIV ab^®^ test from IDEXX^®^, which detects the viral p27 antigen, evidenced a prevalence rate of 23.3% for FeLV in domestic cats in the city of Monteria (14 positives in 60 individuals sampled), which is one of the highest prevalence rates reported for domestic felines worldwide [19]. Notably, although most animals were in good health, the most relevant clinical findings in this population studied were the presence of wounds on the body (13%), weight loss (5%), loss of appetite and decay (5%), and vomiting and diarrhoea (2%) [19].

The gradual increase in the feline population in Colombia and several other countries is accompanied with the appearance of diseases that put the animal’s health at risk. FeLV is one of the main retroviral diseases with higher morbidity and mortality rates in cats in Colombia; therefore, it requires a timely diagnosis that allows for both virus control and life extension in these animals [20]. Therefore, this study represents an approach to the prevalence and genotypification of FeLV in privately owned and shelter cats in the metropolitan area of Aburrá Valley, Colombia.

## 2. Materials and Methods

### 2.1. Type of Study

This cross-sectional descriptive study included samples obtained through a survey, assuming an infinite population, with a 95% confidence interval (CI) and a 10% error. A total of 96 whole blood samples were determined to be required from a proportional number of male and female cats. Considering the possible loss of samples, the sample size was increased to 100 animals. The selection of sample animals was made by convenience based on the sites (privately owned and shelter cats), wherein the owners or managers agreed to participate in the study and signed an informed consent form.

### 2.2. Ethical Considerations

This study was approved by the Animal Research Ethics Committee of the Universidad Cooperativa de Colombia, with headquarters in Bucaramanga (Cert. 004 - November 01, 2016). The owners and managers responsible for the shelters signed an informed consent endorsed by the Ethics Committee. In addition, the authors declare that in the performance of this study, all scientific, technical, and administrative standards for animal research have been observed.

### 2.3. Location

Blood samples were extracted from felines in the Aburrá Valley (Figure 1): Medellín (the core city), Barbosa, Girardota, Copacabana, Bello, Envigado, Itagüí, La Estrella, Sabaneta, and Caldas; Department of Antioquía, Colombia; the average temperature ranges from 22 °C to 13 °C. The bottom of the valley is approximately 1800 m above sea level in Caldas and approximately 1400 m above sea level in Barbosa. It is surrounded by mountains that reach approximately 3000 m above sea level.

### 2.4. Sampling, Collection, and Storage

A cross-sectional study was conducted with the convenience sampling of feline patients attending seven veterinary clinics and belonging to four animal shelters from the metropolitan area of the Aburrá Valley. A cat bag was used to immobilize the cats, which restricts the cat’s movement and facilitates its manipulation, thereby reducing stress to the animal and risks for the sample collector. Subsequently, 1 ml blood was extracted, with a puncture in the cephalic vein, using a 21G gauge needle in a dry tube containing ethylenediaminetetraacetic acid.

Data was collected regarding sex, the approximate age of feline patients, breed, the presence/absence of clinical signs, vaccination and castration. Samples were transported to the laboratory where a rapid enzyme-linked immunosorbent assay (antigen test)—SNAP Combo Ag FeLV / Ab FIV^®^ (IDEXX ™)—was performed for the simultaneous detection of FeLV antigens and antibodies against feline immunodeficiency virus (FIV) in feline serum, plasma or whole blood. The sensitivity (S) and specificity (E) of the SNAP Combo FeLV Ag Diagnostic Test reported by the manufacturer was 98.6% and 98.2%, respectively. Subsequently, the samples were centrifuged at 640× *g* for 15 min to extract the serum, plasma and leukocyte layers, which were stored at −80 °C until further use.

### 2.5. RNA Extraction and Complementary DNA Synthesis

Viral RNA was extracted using the QIAamp^®^ Viral RNA Mini kit (QIAGEN^®^, Hilden, Germany), in accordance with the manufacturer’s instructions. Quality and RNA amounts were determined through spectrophotometry using NanoDrop^®^ ONE (ThermoFisher Scientific^®^, Waltham, Massachusetts, USA), and the aliquots of RNA were stored at −80 °C until further use. For the synthesis of cDNA, the Thermo® Reverse Transcription System kit was used in accordance with the manufacturer’s instructions. A mixture of 1 µL (100 pmol/µL) random hexamer primers and 500 ng total RNA was made, and water was used to adjust to the mixture to 15 µL. RNA was initially denatured at 70 °C for 5 min and immediately incubated on ice. The reverse transcriptase (RT) mixture comprised 5 µL of M-MLV 5× Reaction Buffer (250-mM Tris-HCl, 375-mM KCl, 15-mM MgCl2, 50-mM DTT), 1 µL dNTPs (10 mM), and 0.5 µL M-MLV RT (200 units). This mixture was added to the denatured mixture and reverse transcription was performed in a total volume of 25 µL for 60 min at 37°C in the Proflex PCR system thermal cycler (Applied Biosystems^®^, Foster City, CA, USA).

### 2.6. Polymersase Chain Reaction (PCR) and Sequencing

The clinical samples were subjected to PCR, amplifying a FeLV-U3LTR fragment and gag region, and the Maxima Hot Start PCR Master Mix (2×) (Thermo Scientific^®^, Glen Burnie, MA, USA) was used in accordance with the manufacturer’s instructions. The primers used in PCR and sequencing that amplified a segment of 707 exogenous retrovirus nucleotides were U3-F (1) (5´-ACAGCAGAAGTTTCAAGGCC-3´) y G-R(1) (5´-GACCAGTGATCAAGGGTGAG-3´) [21]. Briefly, 4 µL cDNA was added to the PCR mixture containing 25 µL Maxima Hot Start PCR Master Mix (2×) (Maxima Hot Start Taq DNA polymerase 2×, Hot Start PCR buffer, 400 μM dATP, 400-μM dGTP, 400 μM dCTP, 400 μM dTTP, and 4 mM Mg2 +), 15 µL nuclease-free water and 3 µL (10 µM) of each primer, forward and reverse. The Proflex PCR system (Applied Biosystems^®^, Foster City, CA, USA) was used under the following conditions: initial temperature 95 °C for 4 min, followed by 35 denaturation cycles at 95 °C for 30 s, alignment at 50.8°C for 30 s, extension at 72 °C for 1 min and a final extension at 72°C for 5 min. Ultra-pure water was used as a negative control. Positive samples were also subtyped by conventional PCR for the presence of FeLV-A, FeLV-B and FeLV-C subtypes, using previously described primers [22]. Each reaction comprised a final concentration of 500 nM of each primer, with the PCR mixture containing 25 µL Maxima Hot Start PCR Master Mix (2X) under the following conditions: initial denaturation for 2 min at 98°C followed by 40 cycles of 98°C for 20 s, 52°C (FeLV-C) or 64°C (FeLV-A/FeLV-B) for 30 s and 72°C for 80 s, with a final elongation step of 72 °C for 10 min.

After PCR, 5 µL of each amplicon was analyzed through electrophoresis in a 1.5% agarose gel (AGAROSE I™, Amresco, Solon, OH, USA) at 100 volts for 35 minutes. EZ-VISION ™ dye (Amresco, Solon, OH, USA) was used to stain the gel and the Gel image analyzer Doc TM XR+ (Bio-Rad Laboratories, Hercules, California, USA) was used to visualize the bands. The products were estimated with a molecular weight marker ranging from 100 bp to 3000 bp (GeneRuler™ 100 bp Plus DNA Ladder, Thermo Scientific^®^). Finally, amplicons of the positive samples were sent for sequencing to Macrogen Inc. (Macrogen Inc., Seoul, Korea) for purification and sequencing in ambisense.

### 2.7. Sequencing and Phylogenetic Analysis

The resulting sequences were submitted to the Genbank (MT229928-229957) and compared with sequences deposited in the GenBank database. The sequence edition and assembly were performed using the SeqMan Pro package (DNAStar Lasergene™ V15.0 software package, Madison, Wisconsin, USA). Nucleotide BLAST™ (basic local alignment search tool) was used to explore the sequence identity of FeLV strains obtained with all available FeLV sequences in the NCBI nucleotide databases.

Phylogenetic relationships based on the nucleotide alignment of partial sequences were deduced using distance-based methods (the neighbour-joining algorithm) and character-based methods (maximum likelihood) implemented in MEGA™ 7.0 for Windows®. The best-fit model of nucleotide substitution was identified by MEGA ™ 7.0 and jModeltest. Phylogenetic trees were created using the maximum likelihood method with the Kimura 2 + G model, with a bootstrap of 1000 repetitions.

### 2.8. Statistical Analysis

The data were collected and tabulated in Excel 2010®, according to the variables sampling area, sex, age, and breed. The WinEpi 2.0 software (available online at: http://www.winepi.net/) and Prism® 7.01 for Windows™ (GraphPad Software, La Jolla, CA) were used for the processing and statistical analysis of the data. Descriptive statistics of demographic data were generated. The categorical variables are summarized as frequencies and proportions. A bivariate analysis was performed to compare the positive and negative results using the chi-square and Fisher’s tests according to correspondence. In all cases, P values of <0.05 were considered statistically significant.

## 3. Results

The sampled population was comprised 54% (*n* = 54) female felines and 46% (*n* = 46) male felines, with an average age of 2.35 years, an average weight of 3.0 kg, and a body condition score of 3/5. Overall, 48 were privately owned felines and 52 belonged to shelters. Taking age as the reference, we found that 45% (*n* = 45) of felines were in the range of <1 year, 31% (*n* = 31) in the range of 1–3 years, and 24% (*n* = 24) in the range of >3years (Table 1).

Most of them were clinically healthy, and the owners reported weight loss, gingivitis, and loss of appetite and other neurological signs, while laboratory analyses found anaemia in some feline patients (Table 2). In addition, FeLV was found to be widely distributed in the Aburrá Valley (Table 3).

It was determined that of all the felines sampled, 60% presented a positive result for FeLV through the SNAP Combo Test, indicating an apparent prevalence of 60% (95% CI: 50.40–69.60%) in the feline population assessed. Due to the fact that the used test also diagnosed the presence of antibodies for FIV, three cases were found to be doubly positive (FIV/FeLV), of which the corresponding medical examination determined that one was clinically healthy, another had gingivitis, and another had anaemia at the time of the checking and physical examination. Considering the sensitivity (S) and specificity (E) of the SNAP Combo FeLV Ag Diagnostic Test reported by the manufacturer, it was concluded that the actual prevalence represented 59.44% (95% CI = 49.81–69.06%).

A total of 100% of the feline population included in the study comprised domestic shorthair mixed breeds; FeLV-positive patients belonged to different geographical areas within the Aburrá Valley; therefore, establishing any type of relationship with the distribution of the virus in the city was not possible; age was not a determining factor to detect seropositive patients because the ranges varied from animals that were several months old to up to 14 years old (Table 1); the number of cats in both sex groups was proportional, and it was impossible to establish a relationship between positive patients and FeLV with the clinical signs presented (Table 2), given that more than half were clinically healthy. In addition, no difference could be established between privately owned and shelter cats.

To confirm the diagnosis, U3LTR-GAG RT-PCR was conducted on all samples, of which 30 felines were positive (50% of p27-positive animals). There was a patient that was negative for SNAP IDEXX Ag p27 and positive for RT-PCR, yet the patient did not present any clinical signs at the time of the examination, so was classified as apparently healthy and a false negative patient.

To subtype the FeLV in the samples, we used primers that recognize sequences in the pol gene upstream of the env gene start codon and sequences in the U3 region of the LTR that are conserved among exogenous FeLVs. All samples were classified as FeLV-A; however, two samples were simultaneously positive for FeLV-A and FeLV-B and no samples were positive for FeLV-C.

All the amplified segments of the U3LTR-GAG fragment of 707 nucleotides, which allows for the identification of exogenous FeLV, were sent for sequencing. The samples presented a homology greater than 98% (Table 4). The BLAST analysis found the highest identity with the FeLV-A Glasgow-1 variant. This study compared our sequences with other sequences reported in Brazil, the United States, the United Kingdom, Malaysia and Japan and found that all strains circulating in Aburrá Valley belonged to the FeLV-A subgroup (Figure 2).

In addition, the circulation of a viral variant was evidenced (sample MDE/82/C/2018), which was closely related to the FeLV-FAIDS variant in a feline patient who was clinically healthy under veterinary medical assessment and belonged to the municipality of Bello. As evidenced in the phylogenies presented in Figure 2, some viral variants (MED/61/C/CO/2018 y MED/73/C/CO/2018) were phylogenetically more distant from the FeLV sequence group studied, indicating a high level of viral divergence. One of these patients showed neuropathy signs at the medical examination, and both felines belonged to the municipality of Medellín.

## 4. Discussion

FeLV is a virus that has a great impact on the world’s feline population, and inadequate research at a national level has not allowed for a precise understanding of the viral infection’s epidemiological and molecular dynamics in our country. The present study documented a high prevalence rate of FeLV (59.44%), as revealed by the SNAP Combo FeLV Ag/FIV/Ac test, and a wide distribution in the Aburrá Valley. Although non-probabilistic sampling does not allow for the determination of the real population distribution, the sample population studied herein has one of the highest infection rates reported worldwide [23] and the highest reported infection rate in Colombia to date [19,24].

Recent studies conducted in developed countries, which assessed more than 62,000 cat samples from the United States and Canada at veterinary clinics and shelters [23], reported a general prevalence rate of 3.1%, which may have increased up to 4.7% in animals with oral lesions and up to 8% in animals with respiratory diseases [23]. This prevalence rate is similar to that of Malaysia (1.2%) [25], New Zealand (1–2.6%) [26,27] and Europe, wherein the overall prevalence rate of FeLV in cats visiting a veterinary facility was 2.3% (141/6005; 95% CI: 2.0–2.8%), with the highest prevalence rates in Portugal, Hungary and Italy/Malta (5.7–8.8%) [28], indicating that the prevalence rate of FeLV infection in developed countries is extremely low compared with that which was found in the present study.

Studies conducted in Brazil have reported a wide diversity of positive results depending on the region assessed and the diagnostic methodology used. PCR in the Minas Gerais region revealed an infection rate of 47% in both healthy and infected animals [29]. In the city of Cuiabá, Mato Grosso state, western Brazil, the prevalence rate of infection was 4.5% (95% CI: 1.1–9.1) [30], and in the cities of Ilhéus and Itabuna, in the micro-region of Bahia at the northeast region of Brazil, the reported prevalence rate was 3% (95% CI: 1.1–6.4%) [31].

Similar to the results observed in Brazil, studies in Colombia, although scarcer, have evidenced a huge difference in the distribution of the infection. Studies performed in the 1990s reported prevalence rates between 9% and 12% [17,18], whereas the prevalence rate in Córdoba (in 2009) was 23.3% [19], and in Bogotá it was 13.1% in clinically ill cats [24].

In comparison with the other studies presented in the country, the present study has reported a much higher prevalence rate, indicating a high frequency of FeLV in Aburrá Valley. A possible explanation for this could be the broad transmission mechanism of FeLV, increased FeLV infection by direct contact between individuals with outdoor access and cohabitation and a high population density, which promotes stress and poor hygiene. The bias of the recruitment process of the animals that were brought to veterinary clinics, which could partially justify the high prevalence rate of FeLV found in the present study, cannot be ruled out. However, as reported previously in Brazil [29], more than 50% of the cat population was clinically classified as healthy when brought in for vaccination or routine check-ups (Table 2).

We found that 50% of P27-positive samples were also positive in RT–PCR, which is an indicator of viral infection, given that this method detects exogenous viruses [32]. Conversely, feline patients negative in PCR but positive in SNAP p27 (p27-positive/RNA-negative) were likely to be either presenting an atypical infection, generating intermittent antigenemia, or a low viral load. Similar results have been reported in other latitudes such as in Malaysia, where a difference of 13% in PCR with respect to Snap P27 was found [32] and, in New Zealand, similar to that observed in the present study, 50% of feline patients were reported to be PCR-positive from the total number of P27-positive feline patients [26]. In our study population, we found only one feline patient who was negative to enzyme-linked immunosorbent assay and positive to PCR; therefore, we believe that the possible explanation is that it was a false negative based on the margin of error of the test [33].

Conventional PCR and phylogenetic analyses of the strains in our study revealed that they belonged to FeLV-A subgroup, which is the most abundant one and is responsible for transmission between animals [29]. On the basis of the analysis of the U3LTR region, it is not possible to establish a geographical relationship with the distribution of the circulating virus [32]. A sequence belonging to our study (MDE/82/C/2018) was closely related to the FeLV-FAIDS viral variant, which has been characterized by inducing the development of an immunodeficiency syndrome characterized by persistent antigenemia, decreased circulating CD4+ T lymphocytes, T-cell-dependent immune responses, and opportunistic infections [34,35]. It is necessary to perform subsequent genomic analyses of these genomic variants, as it is widely known that the FeLV-T viral subtype is a virus that originated from the FeLV-FAIDS type viruses in the early 1990s [36,37].

FeLV-A is the major virus infecting cats, and its evolution has created the other subgroups due to mutations on the env gene [7]. In agreement with other studies, all FeLV-B-positive cats were apparently co-infected with FeLV-A. As reported in different areas of the world [29,38,39], it has been shown that the subgroup B viruses can be isolated in infected cats, but only in conjunction with FeLV-A, suggesting that horizontal transmission of FeLV-B alone is an uncommon epidemiological event [29].

It is also important to highlight the clinical relevance of the knowledge of the circulating genetic variants, since a great therapeutic difficulty has been described in feline patients positive for FeLV-FAIDS and FeLV-T variants that are refractory to antiviral treatments [35], which, although not frequently used in our country, could help in improving the quality of life of FeLV-positive patients in the country.

FeLV has a high mutation power, and its molecular characterization provides a complete virus vision; therefore, sequencing other genetic segments of the virus and performing molecular tests with other types of feline samples would help identify other viral subgroups and even targeted infections. As the mechanisms of viral infection, replication, and mutation processes become clearer, medical treatments can be established, aimed at reducing viral loads, inhibiting viral replication, and therefore improving the life quality and expectancy of our patients.

## 5. Conclusions

The Aburrá Valley has one of the highest frequencies of FeLV. For an effective diagnosis of the presence of the virus in our feline patients, commercial tests (AgP27) must be combined with molecular evaluations, considering the fact that it is not possible to relate the virus to a specific clinical sign. It is necessary to continue investigating this virus to learn its pathogenesis and viral replication mechanisms, which can result in effective therapies for the control and elimination of the virus in our feline patients. Further studies are warranted to establish the prevalence of viral infections in felines in Colombia, specifically leukemia, as well as additional reports on the casuistry and distribution of the virus in our country in order to implement effective prevention measures.

## Figures and Tables

**Figure 1 viruses-12-00464-f001:**
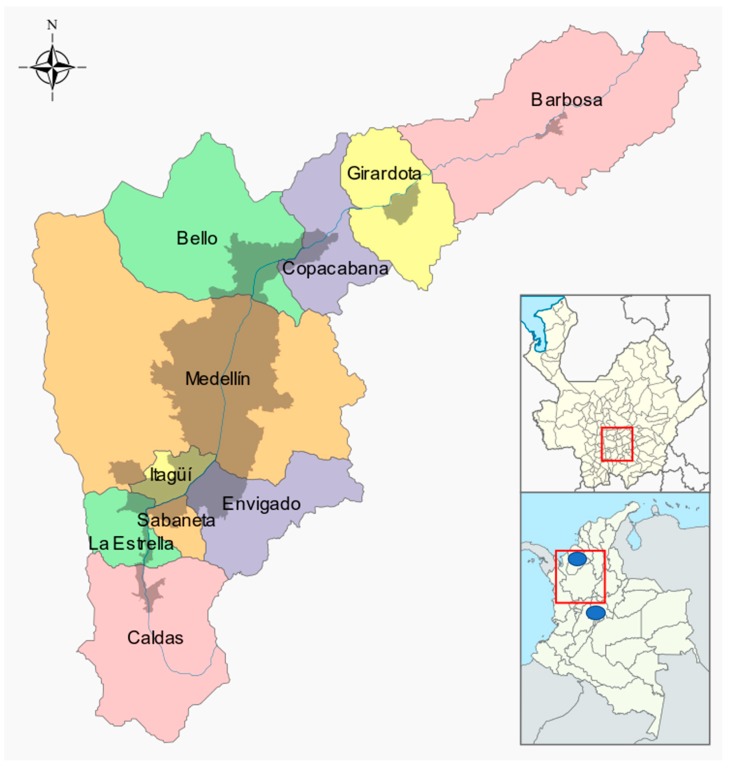
Geographical location of the Aburrá Valley, where the antigenic and molecular presence of FeLV was assessed. Magnification area defined in red box. The grey shadow represents urban areas. The blue dots in lower right panel represent the cities of Monteria and Bogotá. The map was obtained from the official page of the metropolitan area of the Aburrá Valley (CC BY-SA 4.0 license). See text for references.

**Figure 2 viruses-12-00464-f002:**
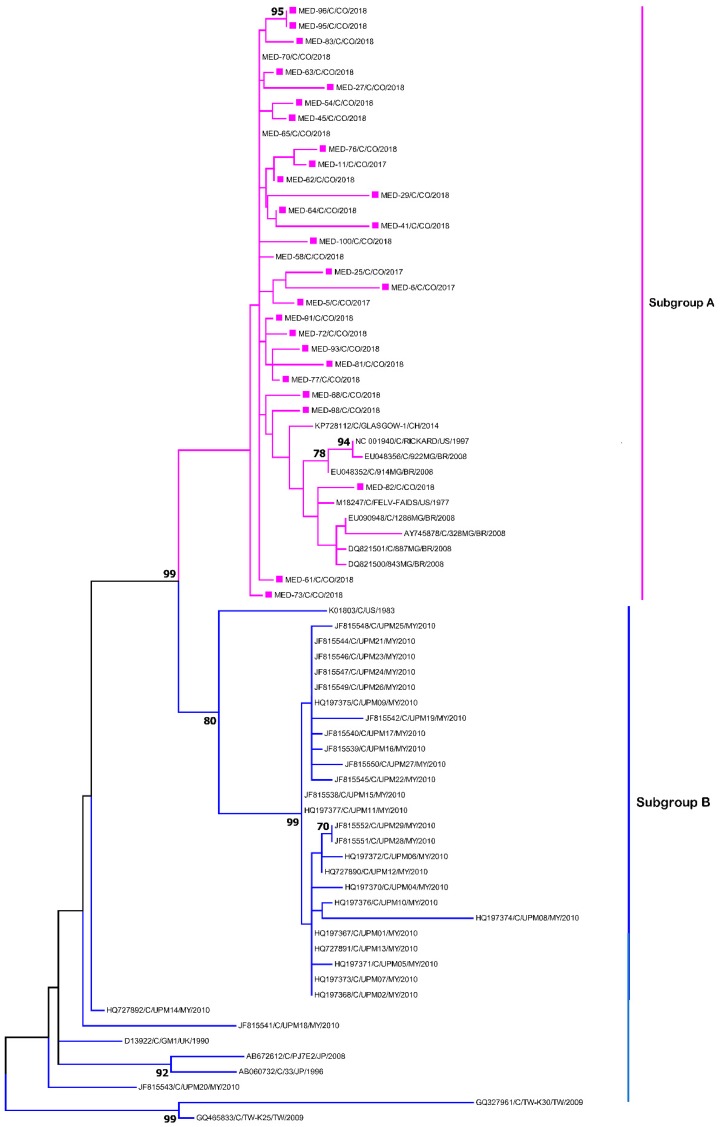
Phylogenetic tree of the U3LTR-GAG segment of feline leukemia virus (FeLV) using the maximum parsimony bootstrap of 1000 repetitions. Pink squares represent Colombian sequences.

**Table 1 viruses-12-00464-t001:** Patients’ age ranges in terms of seropositive results.

Male	Female	AGE Range	Ag FeLV Positive
23	22	<1 years	22
12	19	1–3 years	22
11	13	>3 years	16

**Table 2 viruses-12-00464-t002:** Main clinical signs (Sx) of patients included in the study, with the total of felines that presented the same signs.

Clinic Sx	Patients /%
**A/P Healthy**	54
**Rough Hair**	11
**Anaemia**	3
**Underweight**	8
**Conjunctivitis**	2
**Gingivitis**	8
**Diarrhea**	7
**Mycosis**	1
**Neuropathy**	2
**loss of Appetite**	3
**lymphoma**	1
**Total**	**100**

**Table 3 viruses-12-00464-t003:** Distribution of felines with FeLV positivity in Aburrá Valley, Colombia.

Municipality	Samples	Ag FeLV Positive
Medellin	64	33
Caldas	14	9
Bello	8	6
Envigado	6	6
Itagui	3	2
Santa Helena	3	3
Sabaneta	2	1
	100	60

**Table 4 viruses-12-00464-t004:** Distances (amino acids) from the GAG region of FeLV. The most representative sequences in the Aburrá Valley and their distance with reference strains of the FeLV-A and FeLV-B subgroups are presented.

Subgroup	Strain	1	2	3	4	5	6	7	6	8	10	11	12	13	14	15	16	17	18	19	20	21	22	23
A	**1.MED-68/C/CO/2018**																							
**2. MED-62/C/CO/2018**	0.006																						
**3, MED-73/C/CO/2018**	0.006	0.006																					
**4.MED-61/C/CO/2018**	0.008	0.006	0.008																				
**5. MED-98/C/CO/2018**	0.01	0.008	0.012	0.01																			
**6. MED-96/C/CO/2018**	0.01	0.008	0.01	0.01	0.015																		
**7. MED-25/C/CO/2018**	0.01	0.008	0.01	0.01	0.015	0.012																	
**8.MED-93/C/CO/2018**	0.012	0.01	0.012	0.012	0.017	0.014	0.014																
**9. MED-29/C/CO/2018**	0.014	0.012	0.014	0.014	0.019	0.015	0.015	0.017															
**10.MED-82/C/CO/2018**	0.015	0.014	0.015	0.015	0.017	0.017	0.017	0.019	0.021														
**11. EU048352/C/914MG/BR/2008**	0.014	0.015	0.015	0.017	0.015	0.019	0.019	0.021	0.023	0.017													
**12.M18247/C/FELV-FAIDS/US/1977**	0.014	0.012	0.014	0.014	0.015	0.015	0.015	0.017	0.019	0.01	0.012												
**13. NC_001940/C/RICKARD/US/1997**	0.015	0.017	0.017	0.019	0.017	0.021	0.021	0.019	0.025	0.023	0.006	0.017											
**14. DQ821501/C/887MG/BR/2008**	0.017	0.015	0.017	0.017	0.019	0.019	0.019	0.017	0.023	0.014	0.015	0.008	0.017										
B	**15. HQ197377/C/UPM11/MY/2010**	0.023	0.025	0.025	0.027	0.029	0.029	0.029	0.031	0.033	0.031	0.029	0.033	0.031	0.037									
**16. K01803/C/US/1983**	0.025	0.023	0.025	0.025	0.031	0.027	0.027	0.029	0.031	0.029	0.035	0.031	0.037	0.035	0.027								
**17. HQ197375/C/UPM09/MY/2010**	0.025	0.027	0.027	0.029	0.031	0.031	0.027	0.033	0.035	0.033	0.031	0.035	0.033	0.039	0.002	0.029							
**18. HQ727892/C/UPM14/MY/2010**	0.029	0.027	0.025	0.025	0.035	0.031	0.027	0.033	0.033	0.037	0.039	0.035	0.041	0.039	0.041	0.039	0.042						
**19 JF815550/C/UPM27/MY/2010**	0.031	0.033	0.033	0.035	0.037	0.037	0.033	0.039	0.041	0.039	0.033	0.041	0.035	0.044	0.008	0.035	0.006	0.014					
**20. D13922/C/GM1/UK/1990**	0.041	0.042	0.039	0.041	0.042	0.046	0.042	0.048	0.048	0.044	0.042	0.042	0.048	0.046	0.052	0.054	0.054	0.058	0.06				
**21. AB060732/C/33/JP/1996**	0.051	0.048	0.046	0.046	0.052	0.052	0.052	0.052	0.052	0.058	0.056	0.056	0.058	0.06	0.064	0.06	0.066	0.069	0.068	0.041			
**22. GQ465833/C/TW-K25/TW/2009**	0.06	0.062	0.058	0.06	0.066	0.064	0.064	0.064	0.068	0.071	0.066	0.069	0.064	0.069	0.062	0.069	0.062	0.068	0.068	0.042	0.062		
**23. GQ327961/C/TW-K30/TW/2009**	0.1	0.098	0.098	0.1	0.1	0.1	0.1	0.1	0.1	0.1	0.1	0.1	0.1	0.1	0.01	0.1	100	0.1	0.1	0.085	0,1	0.05

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
