# Peer review of "Prevalence and Genomic Diversity of Feline Leukemia Virus in Privately Owned and Shelter Cats in Aburrá Valley, Colombia"

_viruses, 2020, doi:10.3390/v12040464_

Round 1

Reviewer 1 Report

The presented study deals the prevalence and genotypification of FeLV in house and shelter cats in the Metropolitan Area of the Aburrá Valley, Colombia.

It is not clear, which is the reason FIV results are presented and information about this virus is not included in the different sections of the study (example: title, introduction and discussion etc). If the authors consider incorporating FIV data into the study, it will be highly recommended to generate sequences to identify the viral genotype.

This study is interesting, the design of the study and the protocol appear adapted. From my point of view, however, it is necessary to attend some corrections and suggestions.

In the introduction
In the line 38. Missing order of the viral taxonomy.
In the line 96. “ELISA test” is incorrect, change to “lateral flow immunochromatography assay”.
In the lines 140, 199. Change “race” to “breed”

In the materials and methods section

In L-209. What is the means “the range of 0 to 1 year”. Did you collect samples of cats at birth?
For the detection of FIV, how the authors differentiated maternal antibodies from antibodies generated by the infection?

In the results section

In L-235,239 and 334. Change “symptoms” to “signs”

In L-247. Homology is not equal to similarity. Review the concept.

It is necessary that the sequences generated in the study to be submitted to the GenBank for the assignation of access numbers

What is the reason why amplified products from the env region were not sequenced? The env gene has been described as the adequate region for phylogenetic analyzes. It is very important to validate that the LTR-gag region are genes that can be used for a correct genetic classification of FeLV.

IN THE DISCUSSION SECTION

What was the association of positivity to FeLV with anemia, gingivitis, mycosis and lymphoma? include it in the discussion

Line 305. What does CRP mean?

Is necessary discuss the information obtained from house and shelter cats. It is part of the study title and no data are shown.

In the tables and figures

In the figure 1. Serological identification is a concept incorrect, for FeLV, since the test detects antigen.

In the figure 2. Include bootstrap values in the main branches. Include sequences of genotype C and sequences exogenous viruses in the tree.

In the table 1. Remove in the title “1” “…results 1and molecular…”
"Ag FIV positive” is incorrect, the FIV test detect antibody. revi

In the table 4. The U3 region of the LTR is not coding, review the analysis performed.

In the table 2. How was lymphoma identified? And what was its location?

It is important to search for the association of vaccinated animals and the prevalent genotype found. I could hypothesize about the protection provided by the vaccine

In the conclusions section
Line 331. It is incorrect refer to seropositivity, when the test used detects antigen.

Lines 336-339. It is not a conclusion of the study. The information in this section is very extensive, it is necessary to summarize and improve the sentences of the conclusion.

There are incomplete references, mainly those obtained from Colombia. Revise that they comply with the required format for the journal.

Author Response

RESPONSE TO REVIEWERS COMMENTS (Manuscript ID: viruses-717827)

Prevalence and genomic diversity of the Feline Leukemia Virus (FeLV) in house and shelter cats in the metropolitan area of the Aburrá Valley, Colombia

Carolina Ortega; Alida C. Valencia; July Duque-Valencia; Julián Ruiz-Saenz   

Dear Editor

Please find below our point by point responses to the comments regarding our Manuscript ID: viruses-717827, formerly entitled “Prevalence and genomic diversity of the Feline Leukemia Virus (FeLV) in house and shelter cats in the metropolitan area of the Aburrá Valley, Colombia”. The changes are highlighted in Yellow in the file.

We would like to thank the Reviewers for their helpful suggestions, for critical analysis of the manuscript, and for providing new discussion topics.

Reviewer #1 (Technical Comments to the Author): 

The presented study deals the prevalence and genotypification of FeLV in house and shelter cats in the Metropolitan Area of the Aburrá Valley, Colombia.

It is not clear, which is the reason FIV results are presented and information about this virus is not included in the different sections of the study (example: title, introduction and discussion etc). If the authors consider incorporating FIV data into the study, it will be highly recommended to generate sequences to identify the viral genotype.

R/. We agree to the reviewer. FIV information does not contribute to this article. It was included because it was part of the diagnosis, but it is not relevant. Almost all FIV information was withdrawn because it was not relevant for the development of the work.

This study is interesting, the design of the study and the protocol appear adapted. From my point of view, however, it is necessary to attend some corrections and suggestions.

In the introduction. In the line 38. Missing order of the viral taxonomy.

R/. We agree to the reviewer. The full taxonomy according to the ICTV was added (line 39)

In the line 96. “ELISA test” is incorrect, change to “lateral flow immunochromatography assay”.

R/. We agree to the reviewer. The sentence was corrected

In the lines 140, 199. Change “race” to “breed”

R/. We agree to the reviewer. The sentences were corrected

In the materials and methods section

In L-209. What is the means “the range of 0 to 1 year”. Did you collect samples of cats at birth?

R/. We did not collect samples at birth. Young animals and puppies were classified as Younger than One year. The sentences were corrected

For the detection of FIV, how the authors differentiated maternal antibodies from antibodies generated by the infection?

R/. We did not do that, because FIV is not in the scope of the paper. Almost all FIV data were deleted from the paper.

In the results section

In L-235,239 and 334. Change “symptoms” to “signs”

R/. We agree to the reviewer. The sentences were corrected

In L-247. Homology is not equal to similarity. Review the concept.

R/. We agree to the reviewer. The correct concept that we must use is “Identity”. The text were modified according

It is necessary that the sequences generated in the study to be submitted to the GenBank for the assignation of access numbers

R/. We agree to the reviewer. The sequences were submitted via Bankit

What is the reason why amplified products from the env region were not sequenced? The env gene has been described as the adequate region for phylogenetic analyzes. It is very important to validate that the LTR-gag region are genes that can be used for a correct genetic classification of FeLV.

R/. We agree to the reviewer. However. The specific set of primers used was not designed for Phylogenetic analysis (only for conventional PCR), so does not covers critical areas required for Phylogenetic analysis of the ENV gene.

IN THE DISCUSSION SECTION

What was the association of positivity to FeLV with anemia, gingivitis, mycosis and lymphoma? include it in the discussion

R/. due to descriptive design of the papers and the low numbers of animals showing those clinical signs, we did not performed any statistical association analysis.

Line 305. What does CRP mean?

R/. We apologize for the mistake and correct the sentence. It Was PCR

Is necessary discuss the information obtained from house and shelter cats. It is part of the study title and no data are shown.

R/. We did not find any difference between house and shelter cats. A short sentence was added (Line 239)

In the tables and figures

In the figure 1. Serological identification is a concept incorrect, for FeLV, since the test detects antigen.

R/. We apologize for the mistake and corrected the sentence.

In the figure 2. Include bootstrap values in the main branches. Include sequences of genotype C and sequences exogenous viruses in the tree.

R/. The values in Main branches are included. FeLV-C can not be differenced by sequencing the U3LTR-GAG segment of the virus.

In the table 1. Remove in the title “1” “…results 1and molecular…”

R/. We apologize for the mistake and corrected the sentence.

"Ag FIV positive” is incorrect, the FIV test detect antibody. revi

R/. We apologize for the mistake and corrected the sentence.

In the table 4. The U3 region of the LTR is not coding, review the analysis performed.

R/. The analysis was performed mainly on the GAG amplified portion.

In the table 2. How was lymphoma identified? And what was its location?

R/. The patient was female cat, 11 months old. he presented dyspnea pleural effusion and a mass in left thorax confirmed by echography. Fine needle puncture and cytology showed the presence of lymphoma cells.

It is important to search for the association of vaccinated animals and the prevalent genotype found. I could hypothesize about the protection provided by the vaccine

R/. due to the low vaccination covering in Colombia, no Vaccinated animal were found.

In the conclusions section

Line 331. It is incorrect refer to seropositivity, when the test used detects antigen.

R/. We agree and modify the sentence

Lines 336-339. It is not a conclusion of the study. The information in this section is very extensive, it is necessary to summarize and improve the sentences of the conclusion.

R/. We agree and modify the sentence

There are incomplete references, mainly those obtained from Colombia. Revise that they comply with the required format for the journal.

R/. DONE

Reviewer 2 Report

Firstly I would like to congratulate the authors on their excellent work and fascinating submission. I enjoyed reading the manuscript and would like to see it published.

I have three major suggested areas of improvement, however, before I think it is ready for publication in a high quality journal like Viruses.

  1. READABILITY - Although overall the submission has been well written, there are clunky expressions throughout and choices of words and style that need to be improved. I do not mean to cause offense, but the whole manuscript could be improved if edited by another author who speaks English as their first language, or sent to a paid editing service. E.g. "implied" (L43), "has been accepted" (L46), "it can be affirmed that" (L57), "the combination thereof" (L89), "race" (L140), "and so on" (L141) are all unusual and clunky choices of words/expressions. Unfortunately there are too many things that need improving for a reviewer like myself to go through and individually list them all.
  2. METHODS - I do not think the methodology has been adequately described. E.g. how many of the 100 cats were from the shelter and how many from vet clinics? Which vet clinics were utilised for sampling? How were cats selected for selection in the study? To me this is a very biased population of cats if only 54% were clinically healthy (I know the owners try to argue this L297-298, but I disagree) - which is ok - as long as the methods are adequately described. What is the history of the shelter cats - how long had they been there for, did the cats have any history of outdoor access. etc.?
  3. HIGH RATE OF FALSE-POSITIVE p27 results (30/60, 50%) - I would LOVE the authors to dig a little deeper here and help take this manuscript from an excellent one to an extra-excellent one. I'm frustrated that we just seem to take false-positive p27 results as something that happens without working harder to find out WHY THEY OCCUR. I want someone to work a little harder to solve this mystery for us. Are they truly false-positive p27 results, or are they actually false-negative PCR results due to focal infections or inadequate primer selection? I think there are 2 things the authors could do fairly quickly and easily with residual samples from this study (assuming they are available): (i) contact IDEXX in North America (Missy Bealle) and ask if they will consider running their confirmatory p27 laboratory test for free on the discrepant samples; (ii) send the samples to another laboratory (again IDEXX?) for PCR testing with a different set of primers at a different laboratory. Both of these pursuits would be of great interest to the great FeLV diagnostic conundrum.

MINOR EDITS:

  • Table 1-  is Aburra Valley the dark grey shaded region? If so please describe in the figure legend
  • feline leukaemia virus is always written without capital letters (Title, L12, L34)
  • what is a 'house' cat? Is this a privately owned cat? I suggest renaming to something else (even just pet cat?)
  • Immunochromatography is incorrect (L23) - the SNAP is actually an in-house ELISA test kit
  • L50 - define as FeLV-T in brackets
  • L64 - does this need to be a new paragraph? Suggest all subtype background could be one paragraph
  • L97 insert "viral capsid" p27 antigen
  • L95-103 - it would be helpful for non-Columbians reading this article to know where Bogota and Monteria are in relation to the Aburra Valley. Could this detail be added to Figure 1?
  • L113-114 - suggest rewording "a total of 96 whole blood samples were determined to be required from an equal number of male and female cats"
  • Location (L125-129) - what is the significance of the cats living in a valley? Does this limit cat or owner movement?
  • L142, L269 - Ab for antibody (not Ac)
  • L241-245 -  I was confused by this - if this is a different PCR is should be described in the methods section (2.6). Also please explain further why this was necessary, in addition to the other two PCR assays used?
  • L207-210 - How many shelter cats, how many privately housed cats? Should give age range and also interquartile range. Females = 54%? (not 53.5%?)
  • Table 3 - Heading for last column - FIV meant, not VIF?
  • L226-228 - S and E should be in methods section
  • L241, L320 - ENV should be written env
  • L305 - PCR, not CRP?

Author Response

RESPONSE TO REVIEWERS COMMENTS (Manuscript ID: viruses-717827)

Prevalence and genomic diversity of the Feline Leukemia Virus (FeLV) in house and shelter cats in the metropolitan area of the Aburrá Valley, Colombia

Carolina Ortega; Alida C. Valencia; July Duque-Valencia; Julián Ruiz-Saenz   

Dear Editor

Please find below our point by point responses to the comments regarding our Manuscript ID: viruses-717827, formerly entitled “Prevalence and genomic diversity of the Feline Leukemia Virus (FeLV) in house and shelter cats in the metropolitan area of the Aburrá Valley, Colombia”. The changes are highlighted in Yellow in the file.

We would like to thank the Reviewers for their helpful suggestions, for critical analysis of the manuscript, and for providing new discussion topics.

_________________________________________________

Reviewer #2 (Technical Comments to the Author):

Firstly I would like to congratulate the authors on their excellent work and fascinating submission. I enjoyed reading the manuscript and would like to see it published.

I have three major suggested areas of improvement, however, before I think it is ready for publication in a high quality journal like Viruses.

READABILITY - Although overall the submission has been well written, there are clunky expressions throughout and choices of words and style that need to be improved. I do not mean to cause offense, but the whole manuscript could be improved if edited by another author who speaks English as their first language, or sent to a paid editing service. E.g. "implied" (L43), "has been accepted" (L46), "it can be affirmed that" (L57), "the combination thereof" (L89), "race" (L140), "and so on" (L141) are all unusual and clunky choices of words/expressions. Unfortunately there are too many things that need improving for a reviewer like myself to go through and individually list them all.

R/. The paper was reviewed by ENAGO editing services. See attached certify.

METHODS - I do not think the methodology has been adequately described. E.g. how many of the 100 cats were from the shelter and how many from vet clinics? Which vet clinics were utilised for sampling? How were cats selected for selection in the study? To me this is a very biased population of cats if only 54% were clinically healthy (I know the owners try to argue this L297-298, but I disagree) - which is ok - as long as the methods are adequately described. What is the history of the shelter cats - how long had they been there for, did the cats have any history of outdoor access. etc.?

R/. We partially agree to the reviewer. A short sentence was added to the METHODS section. Due to the data protection policy of our country we cannot identify the veterinary clinics that participate of the Study. We also agree to the bias in sampling due to the Convenience sampling method. Five different shelters participate of the sampling all of the have rescued animals that used to live in outdoors (Stray cats)

HIGH RATE OF FALSE-POSITIVE p27 results (30/60, 50%) - I would LOVE the authors to dig a little deeper here and help take this manuscript from an excellent one to an extra-excellent one. I'm frustrated that we just seem to take false-positive p27 results as something that happens without working harder to find out WHY THEY OCCUR. I want someone to work a little harder to solve this mystery for us. Are they truly false-positive p27 results, or are they actually false-negative PCR results due to focal infections or inadequate primer selection? I think there are 2 things the authors could do fairly quickly and easily with residual samples from this study (assuming they are available): (i) contact IDEXX in North America (Missy Bealle) and ask if they will consider running their confirmatory p27 laboratory test for free on the discrepant samples; (ii) send the samples to another laboratory (again IDEXX?) for PCR testing with a different set of primers at a different laboratory. Both of these pursuits would be of great interest to the great FeLV diagnostic conundrum.

R/. We agree to the reviewer. However in Colombia and for research purposes, we cannot sent animal samples out of the country. There are a lot of regulations that does not allow us to do that.

All negative samples has good quality and quantity by Nanodrop-One. We double check those samples negative samples to confirm the results as has been recommended in previous analysis.

A short sentence was added to the discussion to strong the DISCUSSION section of our paper. We also agree to the reviewer the frustration about discrepant PCR/p27 results. There is an urgent worldwide need of clarify this item, however that was not the aim of our paper.

MINOR EDITS:

Table 1-  is Aburra Valley the dark grey shaded region? If so please describe in the figure legend

R/. WE modified this in the text

feline leukaemia virus is always written without capital letters (Title, L12, L34)

R/. Done

what is a 'house' cat? Is this a privately owned cat? I suggest renaming to something else (even just pet cat?)

R/. We agree and modified in all text.

Immunochromatography is incorrect (L23) - the SNAP is actually an in-house ELISA test kit.

R/. WE modified this in the text

L50 - define as FeLV-T in brackets

R/. WE modified this in the text

L64 - does this need to be a new paragraph? Suggest all subtype background could be one paragraph

R/. WE modified this in the text

L97 insert "viral capsid" p27 antigen

R/. WE modified this in the text

L95-103 - it would be helpful for non-Columbians reading this article to know where Bogota and Monteria are in relation to the Aburra Valley. Could this detail be added to Figure 1?

R/. WE modified this in the figure

L113-114 - suggest rewording "a total of 96 whole blood samples were determined to be required from an equal number of male and female cats"

R/. WE modified this in the text

Location (L125-129) - what is the significance of the cats living in a valley? Does this limit cat or owner movement?

R/. We included this description because th Aburrá Valley is a geographical area that includes multiple municipalities. Our paper represents the data from multiple municipalities, not just one municipality or town as many other previously published papers.

L142, L269 - Ab for antibody (not Ac)

R/. We agree and modified in all text.

L241-245 -  I was confused by this - if this is a different PCR is should be described in the methods section (2.6). Also please explain further why this was necessary, in addition to the other two PCR assays used?

R/. We apologize for the confusion. We modified the text to clarify. To confirm diagnostic, U3LTR-GAG RT-PCR was used. To subtype the FeLV in samples a different conventional RT-PCR  were used  that amplify  recognize sequences in the pol gene upstream of the env gene start codon and sequences in the U3 region of the LTR that are conserved among exogenous FeLVs..

L207-210 - How many shelter cats, how many privately housed cats? Should give age range and also interquartile range. Females = 54%? (not 53.5%?)

R/. Done, included in the text.

Table 3 - Heading for last column - FIV meant, not VIF?.

R/. This column was deleted by another reviewer request.

L226-228 - S and E should be in methods section

R/. Done

L241, L320 - ENV should be written env

R/. Done

L305 - PCR, not CRP?

R/. Done

Round 2

Reviewer 1 Report

Thank you for the revisions

Author Response

RESPONSE TO REVIEWERS COMMENTS (Manuscript ID: viruses-717827)

Prevalence and genomic diversity of the Feline Leukemia Virus (FeLV) in privately owned and shelter cats in the metropolitan area of the Aburrá Valley, Colombia.

Carolina Ortega; Alida C. Valencia; July Duque-Valencia; Julián Ruiz-Saenz   

Dear Editor

Please find below our p response to the comments regarding our Manuscript ID: viruses-717827, formerly entitled “Prevalence and genomic diversity of the Feline Leukemia Virus (FeLV) in privately owned and shelter cats in the metropolitan area of the Aburrá Valley, Colombia”.

Reviewer #1 (Technical Comments to the Author): 

Thank you for the revisions

R/ We would like to thank the Reviewer for their helpful suggestions, for critical analysis of the manuscript, and for providing new discussion topics.

Reviewer 2 Report

As I said before, this is a great study and I want to see it published. The authors wrote a convincing response to address my previous concerns, but I was disappointed to find that some of the things they said had been changed had in fact not been changed. For example:

  • "House" cat still referenced in abstract (instead of privately owned)
  • "immunochromatography" still used L24 and L307 (should be ELISA)
  • "viral leukemia" and "feline leukemia" when should just be FeLV (L222 and 106)
  • Line 27 just make FeLV
  • Feline leukemia virus still has capital letters in L13
  • CRP still written instead of PCR (L307)
  • RT should be defined as reverse transcriptase (RT)

Also, again I am sorry to say but the English writing style is not of high enough quality for a high impact journal like Viruses. I take on board that the authors have used ENAGO English services, but in my opinion there are still many clunky phrases and choice of words or word order that make this clear it has been written by scientists with English as their second language.

I have suggested confidentially to the editors a way to address this issue as I do want to see this study published.

Author Response

RESPONSE TO REVIEWERS COMMENTS (Manuscript ID: viruses-717827)

Prevalence and genomic diversity of the Feline Leukemia Virus (FeLV) in privately owned and shelter cats in the metropolitan area of the Aburrá Valley, Colombia.

Carolina Ortega; Alida C. Valencia; July Duque-Valencia; Julián Ruiz-Saenz   

Dear Editor

Please find below our point by point responses to the comments regarding our Manuscript ID: viruses-717827, formerly entitled “Prevalence and genomic diversity of the Feline Leukemia Virus (FeLV) in privately owned and shelter cats in the metropolitan area of the Aburrá Valley, Colombia”. The changes are highlighted in Yellow in the file.

We would like to thank the Reviewers for their helpful suggestions, for critical analysis of the manuscript, and for providing new discussion topics.

Reviewer #2 (Technical Comments to the Author): 

As I said before, this is a great study and I want to see it published. The authors wrote a convincing response to address my previous concerns, but I was disappointed to find that some of the things they said had been changed had in fact not been changed.

R/. We apologize for the involuntary errors included in the text. We hope that the new version will be totally liked.

"House" cat still referenced in abstract (instead of privately owned)

R/. Corrected

"immunochromatography" still used L24 and L307 (should be ELISA)

R/. Corrected

"viral leukemia" and "feline leukemia" when should just be FeLV (L222 and 106)

R/. Corrected

Line 27 just make FeLV

R/. Corrected

Feline leukemia virus still has capital letters in L13

R/. Corrected

CRP still written instead of PCR (L307)

R/. Corrected

RT should be defined as reverse transcriptase (RT)

R/. Corrected

Also, again I am sorry to say but the English writing style is not of high enough quality for a high impact journal like Viruses. I take on board that the authors have used ENAGO English services, but in my opinion there are still many clunky phrases and choice of words or word order that make this clear it has been written by scientists with English as their second language.

I have suggested confidentially to the editors a way to address this issue as I do want to see this study published.

R/. We are taking  the MDPI English editing service,